# The Role and Regulation of Autophagy and the Proteasome During Aging and Senescence in Plants

**DOI:** 10.3390/genes10040267

**Published:** 2019-04-02

**Authors:** Haojie Wang, Jos H. M. Schippers

**Affiliations:** Institute of Biology I, RWTH Aachen University, 52074 Aachen, Germany; haojielyhn@gmail.com

**Keywords:** aging, senescence, autophagy, proteasome, plants

## Abstract

Aging and senescence in plants has a major impact on agriculture, such as in crop yield, the value of ornamental crops, and the shelf life of vegetables and fruits. Senescence represents the final developmental phase of the leaf and inevitably results in the death of the organ. Still, the process is completely under the control of the plant. Plants use their protein degradation systems to maintain proteostasis and transport or salvage nutrients from senescing organs to develop reproductive parts. Herein, we present an overview of current knowledge about the main protein degradation pathways in plants during senescence: The proteasome and autophagy. Although both pathways degrade proteins, autophagy appears to prevent aging, while the proteasome functions as a positive regulator of senescence.

## 1. Introduction

The growth and development of plants is governed by proteins that act developmental phase-specific to control processes like cell division, differentiation, organ growth, and cell death. The regulation of protein levels relies on the one hand on de novo synthesis and on the other on the regulated breakdown of unwanted or damaged proteins, together referred to as proteostasis. Especially when considering aging and senescence, it is of pivotal importance to remove damaged proteins to maintain cellular integrity. In animals, the accumulation of damaged proteins and the demise of proteostasis is a hallmark of aging [1]. Although aging is often associated with deterioration [2], plant cells fully control and initiate their own senescence and death over the course of the life cycle.

Wear and tear on proteins is in part imposed by the environment in which they act and the biochemical processes they control, which can result in misfolding, aggregation, or mistargeting. As a prerequisite, cells monitor the state of each protein through either chaperones or components of diverse proteolytic pathways [3,4]. Still, potentially the largest fraction of the proteins that are removed through proteostasis systems are actually undamaged. Based on the variable physiological demands of proteins during cellular processes, unwanted proteins are removed. For instance, this is required to control cell proliferation and cell differentiation, which are sequential processes that involve specific protein sets during the course of development. In addition, many proteins have a short half-life, which potentially minimizes the risk of protein damage to prevent the accumulation of aberrant and damaging proteins [5].

Protein turnover is performed mostly through the ubiquitin–proteasome system (UPS) and autophagy. Proteins targeted for degradation by the UPS are labeled with ubiquitin to enable recognition by the proteasome. Subsequently, recognized proteins are unfolded in an ATP-dependent manner and degraded by the proteasome. Whereas the UPS typically removes single proteins, autophagy removes complete protein complexes, protein aggregates, cytosol, and organelles [6]. Still, autophagy only operates inside the cytosol, while the UPS is also present in the nucleus. Considering the differences between UPS and autophagy, one would also expect different functions for both protein clearing systems during plant aging, the onset of leaf senescence, and the progression of senescence. One would expect that the proteasome mainly affects the timing and onset of senescence, while autophagy is highly relevant for bulk protein degradation during the progression of senescence.

## 2. Protein Degradation Pathways

The UPS and autophagy constitute two evolutionarily conserved protein degradation mechanisms in eukaryotic cells. While the proteasome uses its own protease activity to degrade target proteins, autophagy works as an intracellular vesicle trafficking system that moves cargo to specialized vacuolar compartments where digestive proteolysis of the cargo occurs through different types of proteases. Below, we briefly introduce the molecular components of the different systems in plants.

### 2.1. The Ubiquitin–26S Proteasome System

The 26S proteasome is a 2.5 MDa ATP-dependent protease complex composed of two particles, a 19S regulatory particle (RP) and a 20S core particle (CP) (Figure 1) [7]. The internal chamber of the CP houses the protease active sites [8]. The entrance to this chamber is spatially restricted to ensure the entry of only deliberately unfolded and treaded substrates. The recognition and unfolding of substrates is regulated by the RP. The base of the RP contains six AAA-ATPases (Rpt1–6) that control protein unfolding, and three non-ATPase subunits (Rpn1–2 and Rpn10). The lid of the RP consists of eight subunits (Rpn3, Rpn5–9, and Rpn11–12) and is responsible for substrate recognition and deubiquitination [1]. The 26S proteasome specifically recognizes ubiquitinated substrates (Figure 1). Ubiquitination of target proteins is highly controlled and specifically mediated by the concerted action of three enzymes [9]. The E1 activating enzyme covalently binds ubiquitin (UBQ) in an ATP-dependent manner and passes it on to an E2 UBQ conjugating enzyme. The E2 assembles with an E3 UBQ ligase that transfers the UBQ moiety to specific substrate proteins. Since plants contain more than 1000 different E3 ligases [10], it is clear that proteins can be targeted for degradation with surgical precision by the cell. Moreover, this also underlines why the UPS is suitable for the selective degradation of master regulators to guide developmental processes such as, for instance, the onset of senescence.

### 2.2. Autophagy

Autophagy (from ancient Greek, meaning self-eating) is a key intracellular trafficking and degradation pathway that is evolutionarily conserved between eukaryotes [11]. Autophagy functions as a recycling machine to preserve cellular homeostasis and is triggered upon stress and specific developmental phases such as senescence. The initiation of autophagy occurs with the formation of a phagophore (Figure 1), a membrane cisterna that eventually expands to sequester specific cargo, including portions of the cytoplasm, large protein complexes, or organelle parts. The origin of the membrane for the formation of the phagophore has long been undisclosed, but recent findings have suggested that most phagophores are formed using endoplasmic reticulum-derived membrane material [12]. As soon as the growing phagophore is sealed around the target cargo, an autophagosome is formed (Figure 1). The autophagosome matures and eventually fuses with the vacuolar membrane to deliver its cargo for degradation. In principle, two types of autophagy occur, macro- and microautophagy [13]. In macroautophagy, cytoplasmic castoff (such as organelles) are sequestered by a double membrane-bound vesicle and transported to the vacuole. In contrast, microautophagy involves the engulfment of cytoplasm by vacuolar invagination. Here, the macroautophagy pathway is discussed and described.

The formation of an autophagosome and trafficking to the vacuole relies on the activity of autophagy-related (ATG) proteins that are largely conserved between all eukaryotes [14]. The genes that encode the different ATG genes can be divided into five functional classes [15]. The first group of ATG proteins form the ATG1 kinase complex (ATG1, ATG11, ATG13), which in plants is not required for the initiation of phagophores as it is in yeast, but rather acts during autophagosome enclosure [16]. The second group of proteins forms a class III phosphatidylinositol 3-kinase (PI3K) complex (VPS15, VPS34, VPS38, ATG6, and ATG14) that seems to mark the phagophore to distinguish it from other endomembranes in plants [13,17]. ATG9 represents the third class of ATG proteins, and is a transmembrane protein that appears to cycle between endomembranes and the phagophore assembly site (PAS) to ensure nucleation of the phagophore [18]. The class four members ATG2 and ATG18 tether pre-autophagosomal membranes to the endoplasmic reticulum for autophagosome formation with ATG9 in plants [19]. Expansion, autophagosome maturation, and targeting rely on two ubiquitin-like conjugation pathways, that of ATG8 and ATG12 [13]. The conjugation pathways involve the E1 ligase-like ATG7, E2 ligase-like ATG3, and E3 ligase-like ATG12–ATG5–ATG16 complex. ATG8 is a ubiquitin-like protein that is conjugated to phosphatidylethanolamine (creating ATG8-PE). ATG8-PE is found on the phagophore and autophagosome membranes, and is delivered to the vacuole together with the engulfed cargo [14]. In contrast with the previous idea that autophagy is a nonselective protein degradation pathway, it has become apparent that the ATG8 protein serves as a cargo specifier by interacting with so-called cargo-receptors to enable selective autophagy [13]. One of these cargo-receptors is Rpn10, which initiates the targeting of inactive 26S proteasomes for destruction by autophagy in plants [20]. Thus, the UPS and autophagy system are interconnected, suggesting a complex regulation to maintain cellular proteostasis during development and stress.

### 2.3. Proteases

Next to the central protein clearance systems, the plant genome encodes for an additional 800 proteases that process or remove a vast variety of proteins [21]. Here, we do not explicitly discuss proteases, but would like to refer to several interesting observations made in relation to senescence. Proteases can be divided into three types based on the position of the peptide bonds they act on in target proteins; Internal endopeptidases, C-terminal carboxypeptidases, or N-terminal aminopeptidases [21]. Peptidases are classified based on the catalytic mechanism they employ, resulting in six catalytic types: Serine, cysteine, threonine, aspartic, glutamic, and metalloproteases [22,23]. Protease abundance during development is highly regulated, and many accumulate in specific subcellular compartments.

Serine and cysteine proteases are especially strongly upregulated at the expression level during leaf senescence [24,25,26]. SAG12, a marker for the onset of senescence, belongs to the Papain-like cysteine proteases (PLCPs) [27]. In addition, vacuolar processing enzymes (VPEs), another class of cysteine proteases, are also highly expressed during leaf senescence in *Arabidopsis* [28]. Activity-based protein profiling (ABPP) to detect specific protease activities during senescence has shown that only several PLCPs show an increase in activity during senescence, while VPE activity even decreases, despite the increase in transcript abundance [26]. In addition, the activity of SAG12 increases during senescence: However, the *sag12* mutant does not show a visible senescence-related phenotype, indicating that it is dispensable in senescence or acts redundantly to other proteases. Still, accumulating molecular evidence does indicate that SAG12 participates in protein degradation and N remobilization during senescence [29,30]. As autophagy is a protein trafficking system that relies on the proteolytic activity of proteases in the vacuole, it is expected that many senescence-induced proteases might be part of this pathway. In addition, several ATG genes encode proteases, with *ATG4* encoding a cysteine protease that is required for the processing of ATG8. It would be interesting to determine if senescence-associated proteases cooperate with autophagy in the recycling of cellular constituents during senescence. 

## 3. Impact of the Proteasome and Autophagy on Aging and the Onset of Senescence

As main pathways for the degradation of unwanted or misfolded proteins and dysfunctional organelles, the proteasome and autophagy pathways play important roles in the adaption of organisms during stress and ensure their survival and longevity. 

The terms aging and senescence are somewhat differently maintained in human and plant biology [2]. Aging and senescence often refers to a process of deterioration in human biology, while in plant biology these terms do not reflect simply a decline in viability, but rather a distinct developmental phase that may lead to the controlled death of an organ or organism. Senescence is no longer only a hallmark of animal aging, but has recently been uncovered as a positive cellular response that triggers tissue remodeling and minimizes the damage caused by stresses [31,32]. In animals, the activity of the proteasome declines with age [1], resulting in the accumulation of misfolded proteins, which is linked to multiple age-related diseases such as Alzheimer’s, Parkinson’s, or Huntington’s disease. Indeed, the maintenance of proteasome activity with age has been suggested as a key feature in promoting the longevity of naked mole-rats compared to mice [33]. Next to the proteasome, autophagy also promotes longevity in animals [34]. Considering the conservation of both proteolytic pathways, one might expect that the proteasome and autophagy have a similar role in promoting longevity in plants.

In plants, abiotic stress and senescence result in the upregulation of *ATG* gene expression [35], suggesting a specific role for autophagy during these processes. It has been shown that defective autophagy (through the disruption of different ATG genes, including *atg2*, *atg4a/4b*, *atg5*, *atg7*, *atg9*, *atg10*, and *atg18a*) causes early senescence in *Arabidopsis* [36,37,38,39,40,41]. Especially under nutrient-limiting conditions (low nitrate), *atg* mutants display earlier leaf senescence and lower rosette biomass [42,43]. Thus, like in animals, autophagy prevents early aging and the precocious onset of senescence in plants. Recently, this was further demonstrated by the fact that constitutive overexpression of ATG5 or ATG7 results in a significantly delayed onset of leaf senescence [44]. For a long period of time, autophagy has been a synonym for nonselective bulk protein degradation: However, an increasing number of studies have demonstrated the selective autophagic-dependent degradation of cell components, including mitochondria [45], peroxisomes [46], chloroplasts [47], ribosomes [48], the proteasome [20], and the endoplasmic reticulum [49]. During senescence, the delivery of chloroplast material is in part regulated by ATG8-INTERACTING PROTEIN1 (ATI1), a transmembrane protein present in the ER and plastid membrane [50]. The loss of ATI1 causes premature senescence during stress, suggesting that coordinated removal of specific plastid proteins during stress is required to prevent a premature collapse of the complete photosynthetic apparatus. Like plastids, mitochondria are also salvaged during plant development. During senescence, ATG11 is required for mitophagy, the selective breakdown of mitochondria-resident proteins and mitochondrial vesicles [45]. Recently, a new role for ATG8 was found in the regulation of leaf senescence. A multidrug and toxic compound extrusion (MATE) transporter called ABNORMAL SHOOT3 (ABS3) was shown to promote senescence under natural and carbon deprivation conditions in *Arabidopsis* [51]. The senescence-promoting ABS3 pathway functions in concert with the longevity-promoting autophagy pathway to balance plant senescence and survival. ABS3 interacts with ATG8 to promote senescence and protein degradation. Thus, on the one hand, autophagy appears to clearly prevent senescence, but when senescence occurs, autophagy seems important for the controlled salvaging of organelles. This notion is especially underlined by the fact that precocious senescence in *atg* mutants can be prevented by blocking SA-induced cell death [36]. Autophagy counteracts instantaneous cell death by controlling NPR1-dependent salicylic acid (SA) signaling, implying that autophagy has a dual role (executioner and procrastinator) in the onset and progression of senescence to maximize the salvaging of remaining nutrients. 

In contrast with the global upregulation of ATG genes during senescence, only a fraction of the proteasome subunit genes increase their expression [52]. In senescing leaves of oilseed rape and wheat, the proteasome is highly active [29,53], in contrast with the observation that proteasome activity declines with age in animals [1]. Apart from the difference in senescence-induced gene expression, the more interesting point is that knocking down/out subunit genes of the proteasome in *Arabidopsis* often causes a delay in the onset of senescence [54]. Thus, it seems that autophagy and the proteasome influence aging and the onset of senescence differentially in plants. Loss of the regulatory particle subunit RPN10 significantly delays the onset of senescence [54], while the overexpression of RPN5a promotes premature senescence [55]. Knockouts in other subunits are often lethal or cause pleiotropic developmental defects [8]. In addition, the use of proteasome inhibitors is able to delay the onset of senescence [56]. Importantly, the loss of autophagy causes early senescence, which has been linked to SA accumulation [36]. However, as the proteasome is removed by autophagy, a potential accumulation of proteasomes might also cause early senescence [57]. Indeed, proteasome activity is increased in *atg* mutants, indicating that proteasome activity might be implicated in the early senescence phenotype of autophagy mutants, still it might also represent a compensation for the decreased proteolytic activity that occurs.

The proteasome is considered to be a requirement for the degradation of short-lived proteins, which often encompass regulatory proteins, while autophagy remains a bulk protein salvage pathway for more long-lived proteins and protein complexes [58]. Bulk protein degradation is a hallmark of leaf senescence to salvage and remobilize nitrogen and carbon compounds for other developing parts of the plant. This task clearly fits the function of the autophagy pathway. We speculate that the proteasome plays a crucial role in determining the onset of leaf senescence. The specific degradation of regulatory proteins, especially those connected to hormone signaling pathways (see below), has a major impact on the onset of senescence. Still, as autophagy seems to delay senescence, and the proteasome appears to promote senescence, these two pathways need to balance their actions during senescence to optimize nutrient recovery (Figure 2). As indicated above, the number of proteasome complexes is under the control of the autophagy pathway [20]. Interestingly, research in animals has shown that autophagy receptors, which regulate selective autophagy, undergo ubiquitination and thereby become targets of the proteasome [59]. Strikingly, a number of studies have reported that the inhibition of proteasome activity results in the compensatory activation of autophagy [60,61]. Thus, the roles of the proteasome and autophagy pathways during senescence appear to be complex and interconnected. However, we can speculate about the order of events (Figure 2): (i) First, the activity of the proteasome is temporarily enhanced during the onset of senescence to degrade potential negative regulators of senescence, including those of specific hormone pathways; (ii) proteins destined for degradation are initially mainly processed by the proteasome during the onset of senescence; and (iii) cellular dismantling during senescence overloads the proteasome, and more focus is placed on autophagy, the bulk degradation system, to degrade protein complexes and salvage organelles. In addition, autophagy prevents premature programmed cell death (PCD) during the senescence process to ensure that cell death is only initiated during the final phase of leaf senescence.

## 4. Transcriptional Regulation of 26S Proteasome

Interestingly, inactive proteasomes are targeted for degradation by autophagy [20], indicating that the level of the 26S proteasome is adjusted to the demands of the cell. When higher proteasome activity is required, the expression of all core and most accessory factors is coordinately upregulated. In several species, key transcription factors have been identified that control the expression of the whole regulon. In yeast, the C2H2-type zinc-finger transcription factor Rpn4 has been shown to recognize a specific *cis*-element (proteasome-associated control element) found upstream of most proteasome subunit genes [62]. Rpn4 accumulates when proteasome activity is impaired, as it itself represents a protein target with a short half-life [63]. The subsequent activation of proteasome gene expression restores proteasome activity and leads to the degradation of Rpn4, establishing an autoregulatory system to control proteasome activity. In mammalian cells, two Nuclear factor erythroid-derived 2-related factor (Nrf) transcription factors have been shown to control the expression of the proteasome regulon [64,65]. Nrf1 and Nrf2 are basic leucine zipper TFs that were originally found to act during oxidative stress by binding to the antioxidant response element (ARE) in the promoters of their target genes [66,67]. In proteasome subunit promoters, these factors also recognize the ARE motif. Like the yeast transcriptional regulator, both Nrf proteins have short half-lives and are readily degraded. Interestingly, Nrf1 is localized to the endoplasmic reticulum membrane and protected from degradation in its inactive form [68]. Upon proteotoxic stress, Nrf1 is deglycosylated, and its N-terminus is cleaved, resulting in the release of Nrf1 and its translocation to the nucleus. 

In plants, two membrane-bound NO APICAL MERISTEM/ARABIDOPSIS TRANSCRIPTION ACTIVATION FACTOR1/CUP-SHAPED COTYLEDONS2 (NAC) transcription factors have been identified as major regulators of the expression of proteasome subunit genes [69,70,71]. ANAC053 and ANAC078 are able to form a heterodimer and act redundantly to protect the plant from proteotoxic stress. Interestingly, both transcription factors are present in an inactive form in the plasma membrane. The treatment of plants with the senescence-inducing hormone abscisic acid (ABA) or a proteasome inhibitor results in the processing of ANAC053 and the release of its transcriptionally active form to the nucleus [72]. Moreover, during drought stress, *anac053* mutants show a delayed senescence, indicating that the transcriptional regulation of the proteasome is vital to the onset of senescence during stress [72]. Still, an in-depth analysis of the role of both regulators of the proteasome during senescence is so far lacking. To understand if other transcription factors might also control the expression of proteasomal subunit genes in plants, we screened publicly available ChIP-Seq data. Surprisingly, hormone-related transcription factors such as ABA INSENSITIVE 3 (ABI3), ABI5, ETHYLENE INSENSITIVE 3 (EIN3), and BRASSINAZOLE-RESISTANT 2 (BZR2) do not associate with any proteasomal gene, while BZR1 has been found to interact with only 4 out of 55 genes [73,74,75,76,77]. A negative regulator of cell expansion, ILI1 BINDING BHLH 1 (IBH1), associates with 11 proteasomal genes [78]. Overexpression of *IBH1* results in a severe dwarf phenotype and delayed development, including senescence [79]. IBH1 acts antagonistically to PHYTOCHROME-INTERACTING FACTOR4 (PIF4), which is another transcription factor with potentially 7 proteasomal subunit genes as direct downstream targets [78,80]. Moreover, PIFs have been shown to act as positive regulators of senescence [81], fitting with the antagonistic relation to IBH1. Although a number of transcriptional regulators of the proteasome have been identified, it is not known which specifically regulate the expression of the proteasome during the onset of senescence.

## 5. Post-Translational Regulation of the 26S Proteasome

Post-translational modifications (PTMs) play vital roles in cellular processes and can affect protein function, activity, structure, and stability, which increases proteome diversity, complexity, and functionality [82,83]. To date, more than 345 PTMs classified into 11 types have been identified as occurring on different proteasome subunits in yeast [84]. Compared to plants, information on proteasomal PTMs in animals and yeast is much more abundant. In addition, the roles of several proteasome PTMs have been revealed. PTMs affect all major regulatory steps, from the synthesis of the proteasome to its destruction. For instance, phosphorylation of the Rpt6 ATPase subunit affects the assembly of the 26S proteasome in yeast [85]. In addition, phosphorylation of the proteasome by protein kinase A at more than 20 subunits, including the catalytic ones, results in increased proteolytic activity [86]. In addition, phosphorylation impacts substrate affinity, 26S stability, and ATPase activity [84,85,86,87,88]. Other modifications, such as O-GlcNAcylation (O-linked N-acetylglucosamine attachment) of Rpt2, reduce proteasome activity [89], while acetylation on the α6 and β3,6,7 subunits upregulates proteasome activity in mice [90]. The ubiquitin-like domain-containing C-terminal domain phosphatase 1 (UBLCP1) dephosphorylates the Rpt1 subunit to block its ATPase activity and thereby disrupts the assembly of the 26S proteasome in human cells [91]. S-glutathiolation of the free 20Sproteasome not only increases its proteolytic activity but also promotes the entrance of oxidized proteins [92]. Interestingly, relocalization of the proteasome from the nucleolus to the cytoplasm in yeast has been linked to N-myristoylation of the proteasome subunit Rpt2 [93]. Furthermore, aged proteasomes are post-translationally tagged prior to disposal through phosphorylation of the Rpn3 subunit in mice cells [94]. These findings suggest that PTMs might regulate the stability and function of the proteasome during aging.

Until now, the role of PTMs on proteasome subunits in plants has not been well understood. Directed proteome studies of the proteasome in plants have revealed several subunits that can be post-translationally modified [53,95]. Here, we screened the Plant PTM Viewer database [83] to obtain a complete overview of currently detected PTMs on proteasome subunits (Appendix A). In total, 207 PTMs have been detected on proteasomal subunits in *Arabidopsis*, including carbonylation, lysine acetylation, lysine 2-hydroxyisobutyrylation, lysine malonylation, lysine methylation, lysine succinylation, lysine SUMOylation, lysine ubiquitination, methionine oxidation, myristoylation, N-terminal acetylation, N-glycosylation, N-terminus proteolysis, N-terminal ubiquitination, O-GlcNAcylation, phosphorylation, reversible cysteine oxidation, S-Glutathionylation, and S-nitrosylation.

Although the biological significance of these modifications and their roles have hardly been explored, there is an overlap with the PTMs found in yeast, suggesting a possible functional conservation. Proteasome inhibitors are known to cause the ubiquitination of many proteasome subunits, which attracts RPN10, which subsequently can activate proteophagy [20]. In rice, the C2 subunit (PAF2) has been identified as a phosphorylation target of a casein kinase, causing (most likely) nuclear translocation [96].

Phosphorylation of the proteasome might be highly relevant during the onset of senescence, as (for instance) mitogen-activated kinase cascades have been shown to affect the timing of the senescence process in *Arabidopsis* [97]. In all cases, a better understanding of the role of PTMs in the proteasome is needed.

## 6. Interactions Between Phytohormones and the 26S Proteasome

Phytohormones and the 26S proteasome are major regulators during the lifespan of a plant, including in seedling development, maturation, and senescence. Inevitably, positive and negative cross-regulation between them happens. Mass spectrometric analysis has revealed ubiquitylated master regulatory proteins involved in brassinosteroid, auxin, abscisic acid, and ethylene signaling [98], but also for other hormone pathways proteasome-dependent degradation is well known [99,100]. Herein, we focus on the relationship between phytohormones, the 26S proteasome, and senescence.

### 6.1. Ethylene

Ethylene was one of the first hormones known to promote senescence, which it does in an age-dependent manner [101,102]. Upon senescence, the abundance of ethylene-related transcripts increases, while photosynthesis-related transcripts decline [103]. The senescence-promoting effect of ethylene requires so-called age-related changes (ARCs) [101], such as the end of the cell expansion phase, for the leaf to perceive and integrate internal and external signals to activate senescence. The proteasome is known to control both ethylene biosynthesis and signaling by regulating the turnover of key components. Both type 2 and type 3 1-aminocyclopropane-1-carboxylic acid synthase (ACS) proteins, enzymes controlling the rate-limiting steps during the biosynthesis of ethylene, are targeted for degradation by the proteasome [104,105]. The master activator of ethylene responses, ETHYLENE-INSENSITIVE 3 (EIN3), is degraded by the proteasome through the action of EIN3-BINDING F-BOX (EBF) proteins [106]. Thus, proteasome degradation of ACS proteins and EIN3 largely blocks ethylene responses. Still, ethylene signaling is also controlled by the proteasome. Upon ethylene perception, EBF1 and EBF2 are targeted for proteasomal destruction, stabilizing EIN3 [107]. EIN3 has been identified as a senescence-promoting transcription factor [108], while EBF1 and EBF2 are negative regulators of the onset of senescence [109]. Before the onset of senescence, key components of ethylene biogenesis and transcription factors that control ethylene-related genes need to be stabilized, while at the same time negative regulators of ethylene signaling need to be removed by the proteasome. The specificity of the proteasome is largely controlled by E3-ligases, suggesting a shift in E3-ligase activity or substrate recognition during the aging of the plant to activate ethylene signaling.

### 6.2. Abscisic Acid

ABA has a stimulating effect on the onset of senescence [110]. Interestingly, ABA has been shown to inhibit the UPS during germination at high temperatures [111]. Considering the low level of ABA during vegetative growth and the dramatic increase in senescent leaves, it is possible that ABA might control the activity of the UPS during senescence. Still, activation of ABA signaling requires the removal of protein phosphatase 2C (PP2C) proteins such as ABA-INSENSITIVE 1 (ABI1) [112]. ABA-bound PYR/PYL/RCAR ABA receptors (PYLs) can only interact with PP2C proteins in the presence of ABA. The bound PP2Cs are recognized by Plant U-box (PUB) E3 ligases and targeted for proteasomal degradation. The ABA receptors are positive regulators of senescence. Loss of PYL9 delays senescence, while overexpression promotes it [113]. A 12-fold *pyl*/*pyr* knockout has shown dramatic impacts on plant development and ABA-induced senescence [114]. The SENESCENCE-ASSOCIATED E3 UBIQUITIN LIGASE 1 (SAUL1) has been shown to prevent premature ABA-induced senescence [115]. Thus, similarly to ethylene signaling, the turnover of key components in the ABA signaling pathway can either prevent or induce senescence. On the one hand, ABA might restrict proteasome activity, which is certainly worth analyzing during senescence. On the other hand, for ABA-induced senescence, the proteasome needs to remove repressors of ABA signaling to induce downstream senescence-related genes. 

### 6.3. Salicylic Acid

Recently, the regulation of SA signaling through selective protein turnover has been uncovered. Upon SA accumulation, NON-EXPRESSOR OF PATHOGENESIS-RELATED GENES 1 (NPR1) moves into the nucleus, where it acts as a transcriptional coactivator to modulate SA responses [116]. The Cullin-RING LIGASE 3 (CRL3) has long been implicated in controlling SA responses, but no direct interaction with NPR1 has been found [117]. Instead, the SA-binding proteins NPR3 and NPR4 turned out to act as scaffolds for CRL3 to mediate the degradation of NPR1 by the proteasome [118]. NPR4 has a high affinity for SA and only binds NPR1 when it is free of SA. NPR3 has a low affinity for SA and can only interact with NPR1 once it has incorporated SA. Through this mechanism, plants can set the level of NPR1. *Arabidopsis* plants with defects in SA signaling pathways, such as *npr1*, or plants that suffer from low SA levels due to the *NahG* transgene, show delayed yellowing and reduced necrosis during developmental senescence [119]. Interestingly, SA and ABA could antagonistically affect NPR1 protein levels via the CRL3/NPR3/NPR4 module [120]. Whereas SA results in the accumulation of NPR1, ABA results in the degradation of NPR1. Potentially, this mechanism is relevant during leaf senescence, as ABA could repress SA signaling and PCD during the early phases of leaf development by modulating the proteasomal degradation of NPR1.

### 6.4. Auxin

It is well known that the negative regulators of auxin signaling AUXIN/INDOLE-3-ACETIC ACID (Aux/IAA) proteins are degraded by the SCF^TIR1^ complex upon auxin treatment [121,122]. Interestingly, auxin treatment has been shown to suppress the activity of the 26S proteasome by stimulating translocation of a protein named PROTEASOME REGULATOR 1 (PTRE1) from the nucleus to the plasma membrane in *Arabidopsis* [123]. The rapid degradation of AUX/IAA proteins mediated by TIR1 upon auxin treatment is counterbalanced by the simultaneous translocation of PTRE1 from the nucleus to limit the activity of the 26S proteasome. Thus, auxin modulates total nuclear proteasome activity. Auxin flow in plants is controlled by PIN proteins, which show a high turnover rate. Ubiquitylation of PIN2 causes endocytosis from the plasma membrane and subsequent targeting for proteolytic processing [124]. Auxin is known to delay senescence [125,126]: However, the mechanism through which this occurs is so far not known. Potentially, the repression of the activity of the 26S proteasome by auxin might be causal to its delaying effect on senescence.

### 6.5. Cytokinin

The plant hormone cytokinin is, in relation to senescence, mainly known as a stay-green hormone [127,128]. Abiotic stresses such as salt or drought stress result in leaf senescence, which is preceded by a decline in the concentration of active cytokinin [129,130]. Cytokinin signaling, such as ethylene signaling, involves the stabilization of positive regulators and the degradation of negative regulators. In this case, type B ARABIDOPSIS RESPONSE REGULATORS (ARRs) act as promoting factors for cytokinin signaling, while type A ARRs act as negative regulators [131,132]. Interestingly, the loss of the proteasomal subunit RPN12 causes impaired cytokinin responses, indicating a major role of the proteasome in the regulation of cytokinin signaling [133]. During the aging of the leaf and the onset of senescence, an increase in the number of oxidized (carbonylated) proteins is observed [134]. Proteins marked with an oxidative modification are primed for proteolysis. Interestingly, cytokinin inhibits the proteasome-mediated degradation of carbonylated proteins [135]. These findings clearly demonstrate that cytokinin signaling relies on the proteasome, but also that the effect of cytokinin on the proteasome might explain how this hormone can prevent the onset of senescence.

### 6.6. Strigolactone

Mutants in strigolactone signaling were originally identified as late senescence mutants, with the *oresara9* (*ore9)* mutant as a novel F-Box protein in *Arabidopsis* and its orthologue DWARF3 in rice [136,137]. Treatment of plants with a synthetic strigolactone has been shown to promote senescence [138]. In addition, the strigolactone biosynthesis genes *MORE AXILLARY BRANCHES 1*/*3/4* are upregulated in senescent leaves [139]. Furthermore, the senescence-related WRKY53 transcription factor induces the expression of MAX2/ORE9 during leaf aging. ORE9 targets the central brassinosteroid (BR) signal regulators BRASSINAZOLE-RESISTANT 1 (BZR1) and BRI1-EMS-SUPPRESSOR 1 (BES1) for degradation by the proteasome [140]. As BRs repress senescence, the role of ORE9 is to relieve this inhibitory effect. Thus, leaves undergoing senescence not only enhance the level of strigolactone but also recruit the UPS to aid in the perception of the hormone and target negative regulators of senescence for degradation.

### 6.7. Jasmonic Acid

Jasmonic acid (JA) is known as a senescence-promoting plant hormone that is also involved in defense responses and chlorophyll breakdown [141,142,143]. In addition, JA biosynthesis is increased during senescence, as the level of JA during senescence increases at least four-fold [141]. The induction of JA signaling and biosynthesis relies on the removal of jasmonate ZIM-domain (JAZ) proteins in a UPS-dependent manner [144]. JAZ proteins act as repressors of JA responses and have been shown to function as negative regulators of senescence [145,146]. The notion that JA promotes senescence and relies on the removal of JAZ proteins to become active indicates that the proteasome is required for JA-induced leaf senescence.

### 6.8. Gibberellic Acid 

Impaired GA biosynthesis delays senescence, while increased GA responsiveness results in precocious senescence [147]. Plant growth is restricted by DELLA proteins [148]. GA promotes growth by causing the removal of the DELLA proteins via the 26S proteasome pathway. Removal of certain DELLA proteins not only promotes growth, but also results in the activation of senescence-promoting transcription factors of the WRKY family [149,150]. Moreover, overexpression of several DELLA proteins delays senescence [147,149]. Thus, similarly to JA-promoted senescence, the removal of specific DELLA proteins by the 26S proteasome is essential to trigger leaf senescence. Still, several lines of evidence indicate that GA might also inhibit senescence [151,152]. Therefore, the effect of GA on senescence is not completely clarified: However, the removal of DELLA proteins appears to be crucial.

### 6.9. Brassinosteroids

Application of brassinosteroids (BRs) results in a delay of senescence through the action of downstream transcription factors [153,154,155]. The BRASSINOSTEROID INSENSITIVE2 (BIN2) kinase phosphorylates BZR1, resulting in its degradation by the proteasome [156]. BIN2 itself is removed upon BR accumulation, releasing the brakes on BR responses [157]. Not surprisingly, *bin2* mutants display delayed senescence, while *bzr1* mutants show accelerated senescence [153,154]. Thus, the removal of BR-related transcription factors during the aging of the leaf by the proteasome prepares the leaf for senescence [155].

Taken together, the overview presented in this section indicates that the UPS may regulate senescence through its control over multiple hormone signaling pathway components. It would therefore be highly interesting to determine which hormone-related components are removed during the aging of the leaf by the proteasome to promote the onset of leaf senescence.

## 7. Transcriptional Regulation of Autophagy

Initial transcriptome studies on senescing leaves revealed that several senescence-associated genes encode components of the autophagy pathway, indicating that autophagy is activated during senescence [158]. Genes such as *ATG7*, *ATG8a*, *ATG8b*, *ATG8c*, and *ATG9* were among the first identified as being induced during senescence [159]. Mutations in *ATG7* and *ATG9* were already known to cause premature senescence [38]. This might be in part due to an altered sink–source relationship within the leaf, as nutrients are no longer efficiently managed, or it might point out that autophagy prevents the early aging of plants. 

Here, we made use of an extensive RNA-SEQ dataset to monitor the expression of core ATG genes [159]. Among the genes encoding for the ATG1 complex, mainly *ATG1a*, *ATG11*, and *ATG13a* show a strong upregulation during the final phase of leaf senescence (Figure 3). Other genes are already upregulated during the maturation phase and are not further induced during senescence, such as *ATG1c* and *ATG13b*. Most of the genes encoding the phospatidylinositol 3-kinase (PI3K) complex show a steady increase in transcript levels from eight days onwards. Only *ATG14a* shows a strong additional induction during leaf senescence. The *ATG9* complex also displays a clear steady increase in transcript levels with age, whereby its transcript level is further induced during senescence. Genes encoding for the phospatidylinositol 3 phosphate (PI3P) complex subunits ATG2 and ATG18 show a varying transcriptional response during leaf growth. *ATG2* levels rise with age and show an additional induction during the later phases of senescence. Among the genes encoding the ATG18 subunit, only *ATG18a*, *ATG18f*, and *ATG18h* are also induced during the final phase of leaf senescence. The other isoforms are either not induced or reach their maximal expression at the onset of senescence (*ATG18g*). Finally, genes encoding conjugation pathway components follow a largely similar expressional behavior to those of the *ATG18* subunit. However, several genes encoding conjugation components do not show an upregulation in expression during senescence, including the ATG12-conjugating enzyme ATG10, which is essential to the formation of autophagic bodies [40]. In addition, *ATG12b* is downregulated during senescence, while *ATG12a* transcript levels are not induced during senescence. Potentially, the ATG8-directed conjugation pathway might be more prevalent during senescence. That said, *atg10* and *atg12a/b* double mutants display early senescence, especially upon nitrogen starvation [40,160]. Taken together, a general upregulation of ATG genes with leaf age and senescence occurs, implicating the importance of this pathway for the nutrient recovery of senescing leaves.

The strong upregulation of ATG genes during senescence, but also during nutrient starvation, abiotic stress, and biotic stress suggests the existence of demand-driven transcriptional regulation of ATG genes. Still, direct transcriptional regulators of ATG genes appear to be lacking thus far. In animals, the target of rapamycin (TOR) kinase has long been known as a negative regulator of autophagy [161]. Only recently, a similar role for TOR in the regulation of autophagy has been uncovered in plants [162]. Under nutrient-sufficient conditions, TOR is active, while Snf1-related protein kinases (SnRKs) that act as energy sensors are inactive [163]. SnRK1.1 has been shown to activate autophagy in plants, indicating that both central energy-sensing mechanisms modulate cellular autophagy activity in plants [164]. SnRK1.1 inactivates the TOR pathway through the phosphorylation of RAPTOR [165], releasing TOR repression of the autophagy pathway. Furthermore, SnRK1.1 can interact with ATG1a and ATG13a, and it enhances the phosphorylation of ATG1a, suggesting that SnRK1.1 regulates autophagy by modulating the activity of the ATG1 kinase complex [166].

Interestingly, reprogramming by TOR and SnRK kinases occurs in part through the modulation of transcription factor activity. Therefore, it is highly likely that downstream targets of these pathways are direct transcriptional regulators of autophagy. SnRK1 activates basic leucine zipper (bZIP) transcription factors through phosphorylation, such as, for instance, bZIP63, which initiates a low energy response in plants [167]. Next to that, SnRK1.1 is able to interact with the NAC transcription factor ATAF1 [168], which causes early senescence when overexpressed. In contrast, TOR activates BZR1, which represses several NAC transcription factors to repress the onset of senescence [154,155]. Interestingly, stabilization of BZR1 by TOR protects BZR1 from degradation through autophagy but not by the 26S proteasome [169]. Sugar signaling mediated by TOR therefore controls the accumulation of the BR-signaling transcription factor BZR1. The repression of autophagy by TOR might actually be in part mediated by BZR1, as it associates with at least eight ATG genes [75]. Taken together, it can be expected that the transcription factors acting downstream of TOR and SnRK kinases are direct transcriptional regulators of autophagy in plants.

## 8. Interaction Between Autophagy and Hormones

Autophagy is upregulated throughout maturation and senescence, while it is repressed during the growth phase of a leaf. This clear distinction nicely reflects the sink to source transition that occurs during leaf development [170]. Leaves start off as heterotrophic sink tissues that rely on nutrients and resources from other parts of the plant to develop. During the expansion phase, active photosynthesis results in an autotrophic tissue that efficiently assimilates carbon and nitrogen [2,171] and starts to function as a source tissue. Based upon the expressional behavior of the different ATG genes, one would expect that growth-related hormones repress autophagy, while stress and senescence-inducing hormones might activate autophagy. 

Cytokinin is a potent inhibitor of leaf senescence through the maintenance of a sink signature in the tissues where it accumulates [172]. Cytokinin accumulation promotes the activation of extracellular invertases and hexose transporters [173]. Interestingly, genes upregulated in the *atg5* mutant are also upregulated in triple cytokinin receptor mutants [35], suggesting that autophagy is needed for proper cytokinin signaling. However, both *atg* mutants and cytokinin receptor mutants show early senescence phenotypes. Therefore, the transcriptional overlap might also represent an early aging signature, which does not necessarily imply the interaction of both pathways. As indicated above, the growth-promoting BR hormone appears to repress autophagy during the growth phase, but loses this control with the age of the leaf as its levels decline [155]. Still, reports in tomatoes have indicated that BR can also promote autophagy in leaves [174]. Therefore, the exact interaction between BR and autophagy requires further studies to resolve when BR represses or activates autophagy during leaf development.

Interestingly, even under nutrient-sufficient conditions, *atg* mutants undergo early senescence [36]. A phytohormone analysis of *atg5* plants has revealed that they accumulate salicylic acid and to a minor extent jasmonic acid, which are both known to promote senescence [102]. The introduction of the *NahG* transgene into the *atg2* or *atg5* mutant completely suppresses the early senescence phenotype, indicating that this phenotype is related to the accumulation of SA [36]. In contrast, the generation of double mutants with JA or ethylene signaling mutants do not block the early senescence phenotype of *atg2* and *atg5*, indicating that these hormone signaling pathways are not required for the initiation of senescence in *atg* mutants. Other *atg* mutants (*atg7* and *atg18a*) also display increased JA levels and increased expression of the JA-regulated PDF1.2 defensin gene under control conditions [175]. However, upon infection with *Botrytis*, no further induction has been observed, and the *atg* mutants display impaired pathogen resistance. In addition to SA and JA, a link between autophagy and ABA has also been established. A FYVE domain protein required for endosomal sorting 1 (FREE1), which acts as a phosphatidylinositol-3-phosphate-binding protein, and VSP23 have been shown to interact with ABA receptor proteins [176,177]. Thus, sensitivity toward ABA in plants is modulated by both the proteasome and autophagy pathways. It still remains to be analyzed if the activation of ABA responses during senescence depends on the activity of the proteasome or that of autophagy.

## 9. Perspectives

The described opposite roles of autophagy and the proteasome during leaf aging and the onset of senescence indicates that plants, in contrast with animals, differentially use their protein degradation machineries during aging. Autophagy is clearly required for delaying the onset of PCD to allow for efficient nutrient recovery of senescing leaves. Still, how autophagy is regulated during senescence at the transcriptional and post-transcriptional level is largely unknown. While autophagy prevents precocious leaf senescence, mutations in proteasome subunits result in a delay in the onset of senescence. The delay can be explained by the idea that the proteasome is required to remove negative regulators of senescence, such as, for instance, hormone signaling components. However, the specific proteins targeted by the proteasome to initiate senescence are unknown, although they might be components of the different hormone pathways, as indicated above. The identification of such targets and transcriptional and post-transcriptional regulators of the proteasome and autophagy pathways are expected to enable the targeted manipulation of senescence processes in plants. Such knowledge will allow for the breeding of crops with improved senescence properties to boost their yield in the field.

## Figures and Tables

**Figure 1 genes-10-00267-f001:**
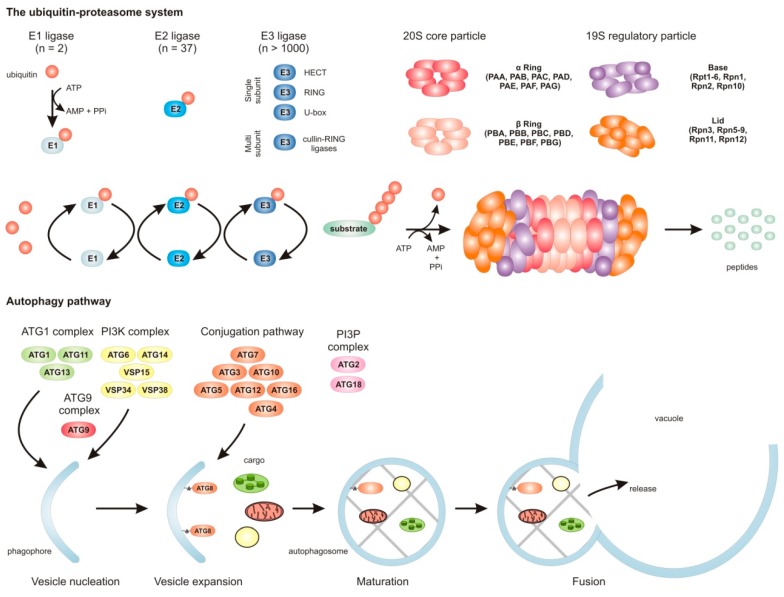
Overview of the main cellular protein clearance mechanisms, the ubiquitin–proteasome system (UPS), and the autophagy pathway. Protein degradation by the UPS is initiated by the specific labeling of target proteins with ubiquitin. Attachment of a ubiquitin molecule requires the action of three enzymes, an ATP-dependent ubiquitin-activating enzyme (E1), a subsequent ubiquitin-conjugating enzyme (E2), and finally a ubiquitin ligase (E3) that transfers the ubiquitin from E2 to a target protein. After (poly)ubiquitination, the target protein is recognized and degraded by the 26S proteasome. Proteasomes contain a 19S regulatory particle and a catalytic active core particle (20S). The 19S regulatory particle recognizes the substrates, deubiquitinates the substrates, and unfolds them at the expense of ATP. The unfolded protein is translocated into the active chamber of the 20S particle to be degraded by the different proteases. The lower part of the figure represents the autophagy pathway. Autophagosomes are initiated by the formation of a phagophore at the outer surface of the ER through the action of the ATG1 complex, ATG9, and the PI3K complex. The growing phagophore surrounds cellular targets in either a selective or nonselective manner. Especially ATG8 plays a major role in substrate recognition for both the selective and nonselective pathways. A mature autophagosome moves toward the vacuole and fuses with it to release its cargo for proteolytic processing.

**Figure 2 genes-10-00267-f002:**
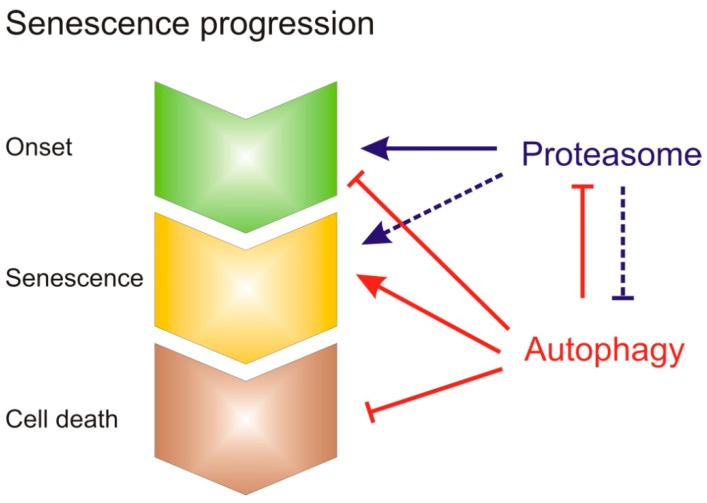
Model depicting the role of the 26S proteasome and autophagy pathways during the onset and progression of senescence. The proteasome acts as a positive regulator of the onset of senescence, while autophagy appears to act as a negative regulator. During the senescence phase, the autophagy pathway is essential for nutrient recovery and the prevention of precocious cell death. Potentially, the proteasome and autophagy pathways affect each other’s functions.

**Figure 3 genes-10-00267-f003:**
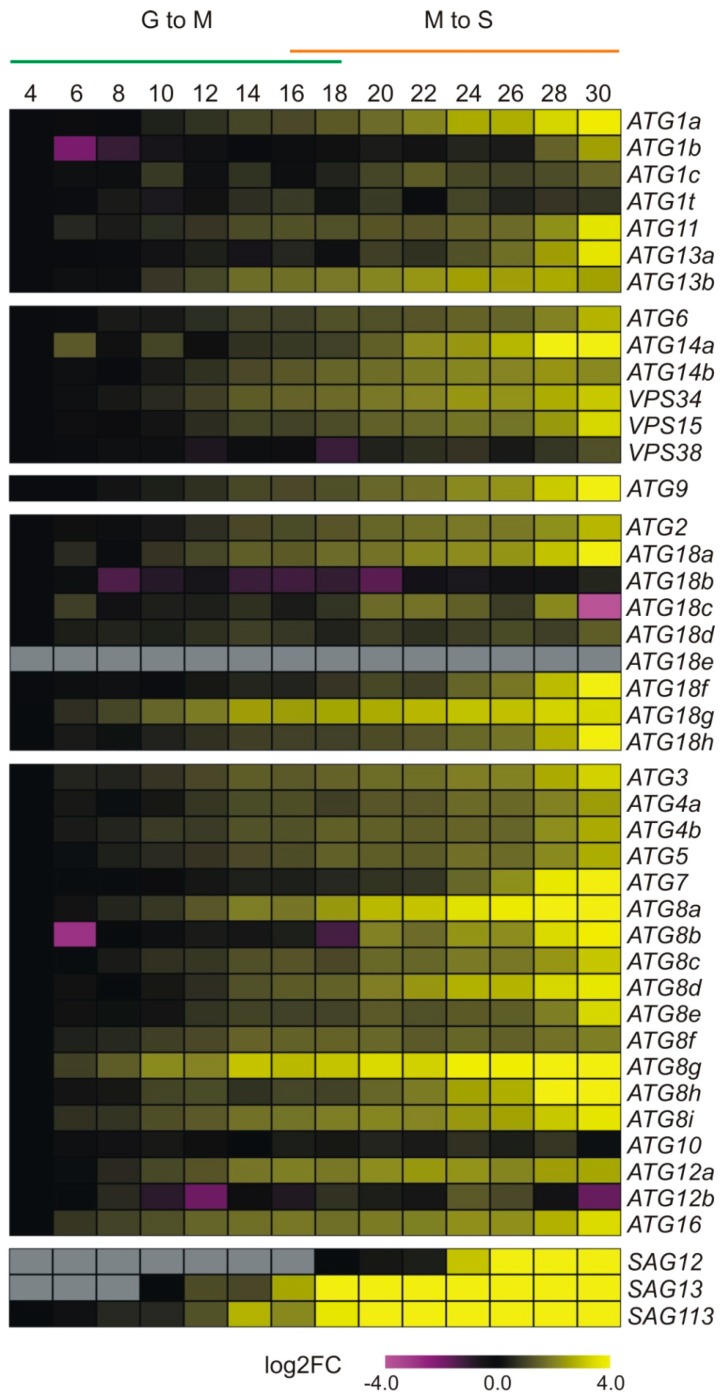
Expression pattern of ATG genes during leaf development. Data from a previous transcriptome analysis (on the fourth rosette leaf at 2-day (d) intervals from 4 to 30 d after emergence from *Arabidopsis*) were used [159]. The leaf lifespan was divided into two stages, growth-to-maturation (G-to-M) and maturation-to-senescence (M-to-S). Expression data are represented as a heatmap based on the log2FC change in Fragments Per Kilobase Million (FPKM) values compared to day 4 for each transcript. Next to the ATG genes, three SAG genes are also shown as representative markers for the onset of senescence.

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
