# Peer review of "The Role and Regulation of Autophagy and the Proteasome During Aging and Senescence in Plants"

_genes, 2019, doi:10.3390/genes10040267_

Round 1

Reviewer 1 Report

This exhaustive manuscript provide a review on the roles and interplay of proteasome and autophagy during senescence. I have several general and more specific comments

from a general point of view this manuscript contain redundancies that can be avoided (see below for details). it also somewhat lacks of focus. many of the experiments described do not have a clear connection with the title and focus of this review, and could therefore be omitted. Some of the hypothese , such as the attractive hypothesis of the sequential role of proteasome and autophagy during senescence will benefit from some more experimental back-up from the published literature.  The interconnection between proteasome and autophagy might benefit from a dedicated subchapter. this will also help to avoid redundancy. The subchapter on proteases can also be moved at the end of the chapter, for clarity. Since the type of bioinformatic analyses conducted by the authors are not described and the criteria, softwares, statistic analyses used are not mentioned, the results and conclusions made from these analyses were not reviewed. 

detail comments follow.

Lines 37-39 this information is too specific at the stage and can be omitted

To avoid redundancy, lines 40-49 can be integrated with lines 51-56 and could be moved either to the introduction or under "2.Protein degradation pathways" 

lines 51-58 specify  which organism(s) are the authors referring to while describing the proteasome  structure?

lines 92-105 which organism(s) are the authors referring to while describing the autophagy mechanism? 

lines 161-173 these arguments are similar to other concepts presented in the introduction and can be integrated

line 174. this is redundant with the information provided in the subchapter of transcription of autophagy

line 179. this seems ti be in contrast with the idea presented in figure 2. can the authors explain the apparent paradox?

line 182 the concept of autophagy once known as a non selective  bulk protein degradation system is described multiple times throughout the review (for example also in line 220), while it could be mentioned just once.

lines 204-205 can be moved to the subchapter "transcription regulation of the proteasome". reference 51 however, do not seem to contain information on the transcription regulation of proteasome subunit genes during senescence. 

line 209, line 213 214. and 229 a reference should be added

lines 233-241 this hypotheses would some sort of experimental support even if taken from the current literature, such as some fine differences in the phenotypical  defects of phenotype of autophagy and proteasome mutants (if there is any).

line 251 this should be discussed together with the content of lines 228-229 (number of proteasomes), possibly in a dedicated subchapter

lines 255-269 this part could be shortened 

lines 277-279 a reference should be added

Chapter 5 as there does not seem to be any clear indication of a role played by post-translation modifications proteasome subunits during senescence, it can be deleted

Chapter 6 as a general rule of thumb it might be easier for the reader if any detail which is not related to proteasome or to senescence could be omitted. 

line 377 a reference should be added

line 379 the function of ACS in ethylene biosynthesis should be mentioned

lines 404-408 these lines can be edited for clarity

lines 434 it would be more appropriate to describe auxin function in terms of " fine tuning" instead of "negative regulator"

lines 442-454 can be edited for clarity

lines 501-503 a reference should be added

lines 529-562 this part could be shortened and edited for clarity to make its connection with senescence more evident  

Author Response

Dear Reviewer,

We thank you for your critical assesment of the manuscript and uploaded a revised version of the manuscript. For detailed responses to your comment, please see the attached response file.

Reviewer 2 Report

Review of  a manuscript entitled 'The role and regulation of autophagy and the proteasome during aging and senescence in plants' by H. Wang and J.H.M. Schippers.

This review is timely, well-organized, and should be of considerable interest to the community (especially and also with respect to the extensive reference list). My comments, which may help to further improve the manuscript, are included in the attached pdf document. One additional comment -- in Figure 3, it may be useful to add gene expression levels for UPS components, e.g., E1, E2, a small selection of E3s (from the hormonal pathways discussed). As the figure shows a pretty complete lifespan, this could nicely compare and contrast some aspects of the UPS and autophagy.

Author Response

Dear Reviewer,

We would like to thank you for your constructive review of our manuscript. We made all adjustments to improve our manuscript and uploaded a revised version. Please find our response file attached.

Round 2

Reviewer 1 Report

This revised version still raise many doubts especially in terms of the experimental back up to some of the hypothesis as I have already highlighted on my first review. In addition the revised version still lacks of focus. The legend to figure 3  lacks details and unfortunately I could not double check with the original data since the reference number 158 cited in the legend text refers to a paper that does not seem to contain transcriptomic analyses ( Peng, P.; Yan, Z.; Zhu, Y.; Li, J. Regulation of the Arabidopsis GSK3-like kinase BRASSINOSTEROID-INSENSITIVE 2 through proteasome-mediated protein degradation. Mol Plant. 2008, 1, 338–346); also  unfortunately  I could not find the Woo et al, plant physiology citation mentioned by the authors in their reply . This did not allow me to evaluate the data contained in figure 3 even in this version of the manuscript.